# Low immunogenicity of common cancer hot spot mutations resulting in false immunogenic selection signals

**Arne Claeys**[1], **Tom Luijts**[1], **Kathleen Marchal**[2,3], **Jimmy Van den Eynden**[1] *

**1** Department of Human Structure and Repair, Anatomy and Embryology Unit, Ghent University, Ghent, Belgium, **2** Department of Information Technology, IDLab, Ghent University, Ghent, Belgium, **3** Department of Plant Biotechnology and Bioinformatics, Ghent University, Ghent, Belgium

* jimmy.vandeneynden@ugent.be

**Data Availability Statement:** This study is based on public data from The Cancer Genome Atlas Network. Code and downstream data used to produce the results described in this manuscript is

## Abstract

Cancer is driven by somatic mutations that result in a cellular fitness advantage. This selective advantage is expected to be counterbalanced by the immune system when these driver mutations simultaneously lead to the generation of neoantigens, novel peptides that are presented at the cancer cell membrane via HLA molecules from the MHC complex. The presentability of these peptides is determined by a patient's MHC genotype and it has been suggested that this results in MHC genotype-specific restrictions of the oncogenic mutational landscape. Here, we generated a set of virtual patients, each with an identical and prototypical MHC genotype, and show that the earlier reported HLA affinity differences between observed and unobserved mutations are unrelated to MHC genotype variation. We demonstrate how these differences are secondary to high frequencies of 13 hot spot driver mutations in 6 different genes. Several oncogenic mechanisms were identified that lower the peptides' HLA affinity, including phospho-mimicking substitutions in *BRAF*, destabilizing tyrosine mutations in *TP53* and glycine-rich mutational contexts in the GTP-binding *KRAS* domain. In line with our earlier findings, our results emphasize that HLA affinity predictions are easily misinterpreted when studying immunogenic selection processes.

## Author summary

The diagnosis of a malignant tumor is preceded by a long process of tumor evolution, characterized by the gradual accumulation of driver mutations, genomic alterations that give cancer cells their typical growth advantage. This process is controlled by the immune system and understanding tumor-immune interactions is critical for the development of new anti-cancer therapies. Immune cells mainly respond to neoantigens, small mutated peptides that are presented at the cancer cell membrane by binding to the Major Histocompatibility Complex (MHC). MHC genes are highly variable between individuals and it was recently suggested that an individual's MHC genotype determines cancer susceptibility. In this study we used a computational approach and simulated a set of virtual cancer patients, based on the expected driver mutation frequencies and each with a similar prototypical MHC genotype. Using these simulations, we show that the earlier perceived

available on GitHub at https://github.com/CCGGlab/mhc_driver and Zenodo at https://doi.org/10.5281/zenodo.4419692.

**Funding:** This work was supported by the Ghent University Special Research Fund Starting Grant (JVdE, BOF.STG.2019.0073.01, https://www.ugent.be/en/research/funding/bof) and Fonds Wetenschappelijk Onderzoek-Vlaanderen (FWO; KM, 3G045620). The funders had no role in study design, data collection and analysis, decision to publish, or preparation of the manuscript.

**Competing interests:** The authors have declared that no competing interests exist.

signals are unrelated to the underlying MHC genotypes, but rather secondary to high frequencies of 13 driver mutations in 6 cancer genes.

## Introduction

Cancer is caused by the gradual accumulation of somatic genome alterations (e.g. point mutations, indels or larger structural alterations) that result in a cellular fitness advantage, positive selection, clonal expansion and eventually malignant tumor formation [1,2]. Somatic point mutations frequently target specific hot spot positions in cancer driver genes, i.e. nucleotides where substitutions lead to a profound functional change in the translated proteins [3]. Many of these driver mutations have now been characterized by large cancer genome sequencing initiatives. Well-studied examples include *BRAF V600E* in melanoma, *IDH1 R132H* in low-grade glioma and *KRAS G12D* in pancreatic and colorectal cancer [4–7].

Somatic mutations also lead to the formation of neoantigens, small novel peptides that are presented to cytotoxic T lymphocytes at the cancer cell membrane by HLA molecules from the Major Histocompatibility Complex type I (MHC-I). Boosting the resulting immune-induced cancer cell killing is the aim of several successful immunotherapies that have been developed in the past decade [8–10]. An additional modulating role of tumor neoantigen presentation by MHC-II is becoming increasingly appreciated [11,12]. The capacity of a neoantigen to bind to HLA molecules from either MHC-I or MHC-II is determined by the mutated peptide's amino acid sequence (8 to 10-mers for MHC-I and larger 13 to 25-mers for MHC-II) as well as the MHC genotype [13]. The MHC-I and MHC-II genotype is each determined by 3 highly polymorphic HLA gene regions on chromosome 6 (MHC-I: HLA-A, -B and -C; MHC-II: HLA-DP, -DQ and -DR), implying thousands of possible MHC genotypes in a population [14].

Several algorithms have been developed to predict the HLA binding capacity of a cancer cell's mutated peptides, based on the peptide sequence and the MHC genotype [15,16]. Both properties can be derived from Next Generation Sequencing data, resulting in the extensive use of HLA affinity predictions to study immunogenic selection processes in cancer [17–19]. However, we recently showed that unexpected associations between mutational sequence contexts and amino acid compositions of HLA binding peptides can easily lead to misinterpretations of HLA affinities [20]. After proper correction for these confounding factors, we found that negative immunogenic selection (i.e. the selective loss of neoantigenic mutations during tumor evolution) is largely undetectable based on HLA affinity predictions.

These conclusions seem to contradict the suggestion that MHC-I and MHC-II genotypes have a profound influence on the oncogenic mutational landscape, i.e. that driver mutations such as *BRAF V600E* only occur in patients where the genotype results in poor presentation of the mutated peptides [21,22]. Here, we generated a set of isogenotypic virtual patients and show that this poor peptide presentability is not caused by any MHC genotype-related selective pressure. We show how the earlier perceived false immune selection signal is caused by 13 lowly immunogenic, common hot spot mutations in 6 cancer genes and identify several functional mutational properties that explain their peptides' weaker-than-average HLA binding properties.

## Results

### Differential HLA affinities between peptides translated from observed and unobserved driver mutations

The initial suggestion that MHC genotypes determine which driver mutations are selected in an individual tumor were largely based on the observation that observed driver mutations

have lower HLA binding capacity than unobserved mutations [21,22]. We first aimed to reproduce these findings.

We selected a set of 688 recurrent missense mutations in cancer driver genes. For each mutation we determined their presence/absence in 10,295 tumors from The Cancer Genome Atlas (TCGA), resulting in a binary mutation matrix (Fig 1A). As expected, the most frequent driver mutation was *BRAF V600E* (5.7%), followed by *IDH1 R132H* (3.8%), *PIK3CA E545K* (2.7%), *PIK3CA H1047R* (2.5%) and *KRAS G12D* (2.0%). Most mutations were relatively rare (median frequency 0.06%), with 675/688 mutations occurring in less than 1% of all samples. The higher mutation frequencies were mainly related to *BRAF V600E* mutations in thyroid cancer (61.8%) and melanoma (43.6%) and *IDH1 R132H* mutations in low-grade glioma (70.1%; Fig 1B). For each mutation and patient, we quantified the HLA affinity using the

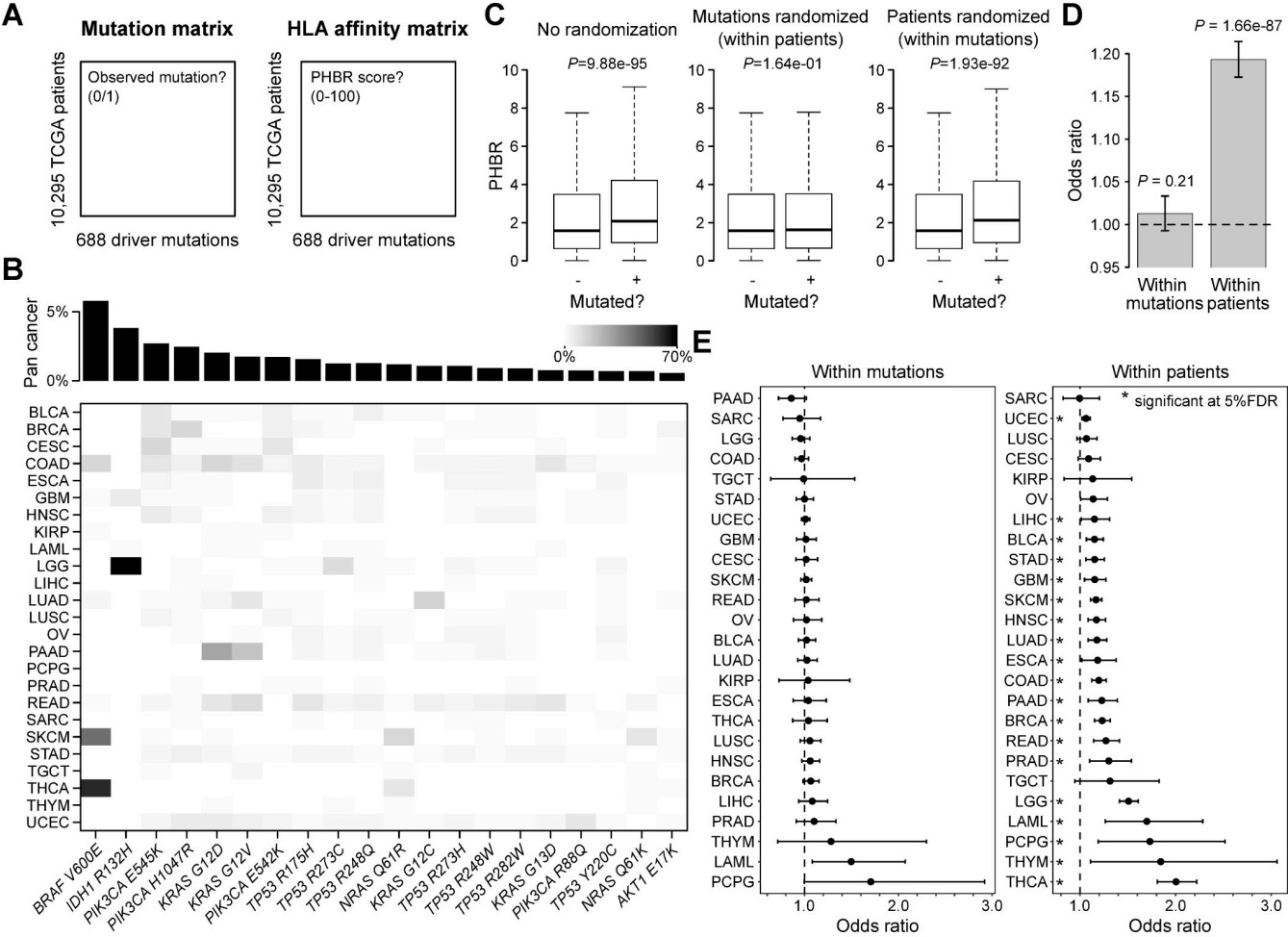

**Fig 1. TCGA driver mutation and HLA affinity analysis.** (A) Illustration of the 2 matrices underlying the study's analytical approach. Driver hot spot missense mutations were identified in TCGA data (see *Methods*) and a *patient x mutation* matrix was generated with each cell containing a binary value (mutation observed or unobserved). A corresponding HLA affinity matrix was then calculated with values representing PHBR values. (B) Mutation frequencies for the 20 most frequent driver mutations. Grey scale shown in top right. Pan cancer mutation frequencies indicated by bar plot on top. (C) Box plots compare PHBR values between observed (+) and unobserved (-) mutations with or without randomization of patients/mutations as indicated. Box plots indicate median values and lower/upper quartiles with whiskers extending to 1.5 times the interquartile range. *P* values calculated using Wilcoxon rank-sum test. (D-E) Pan cancer (D) and per cancer (E) logistic regression analysis between log PHBR (observed variable) and mutation status (0/1, response variable). Analysis was performed both using the within-patient (rows of the matrices) and within-mutation (columns) regression model as indicated (see *Methods* for details). Bar plot (D) and dots (E) show odds ratios with 95% confidence intervals. Asterisks indicate significance at 5% FDR.

previously developed Patient Harmonic Best Rank (PHBR) score (higher affinity = lower PHBR [22]; see *Methods* and S1 Fig), resulting in a second, HLA affinity matrix (Fig 1A).

When comparing PHBR between observed and unobserved mutations (i.e. with the corresponding mutation present or absent in the mutation matrix), we found higher scores for observed mutations (median PHBR 2.08 versus 1.58, $P$ = 9.88e-95, Wilcoxon rank-sum test), at first sight consistent with immune selection acting on neoantigenic peptides. However, while a within-patient randomization of mutations (rows in the mutation matrix) removed this difference ($P$ = 0.16), a similar PHBR difference was still observed after a within-mutation randomization of patients (columns of the mutation matrix; $P$ = 1.93e-92). This observation is a first argument against selection underlying these differences. Indeed, if selection would result in the removal of HLA-presentable mutated peptides, shuffling the patients should remove this effect.

As an alternative approach to quantify the PHBR difference between observed and unobserved mutations, we determined the association between mutation probability and PHBR scores using a logistic regression approach as suggested previously [22]. Two different models were used: 1) to determine how mutation probability correlates to PHBR scores within patients (rows of the mutation matrix), a within-patient model was used that controlled for mutational burden variation between patients; 2) to determine how mutation probability correlated to PHBR scores across patients (columns of the mutation matrix), a within-mutation model was used that controlled for mutation frequency variation between mutations. In line with the randomization approach, a positive correlation was found between mutations and PHBR scores when using the within-patient model (odds ratio (OR) = 1.19, $P$ = 1.66e-87), but not when using the within-mutation model (OR = 1.01, $P$ = 0.21; Fig 1D). Again, these findings are not in agreement with selection processes, which are expected to lead to comparable results when using both models.

Overall, these results strongly argue against selection (at the MHC genotype level) being the reason for differential PHBR scores between observed and unobserved mutations. As this PHBR difference is removed when shuffling mutations within patients and is observed using a within-patient but not a within-mutation model, our findings suggest that mutation frequency is somehow responsible for PHBR differences. In this regard it is remarkable that these differences are more pronounced in individual cancer types with high frequencies of *BRAF V600E* (e.g. thyroid cancer, THCA) or *IDH R132H* (e.g. low-grade glioma, LGG; Fig 1E).

### Weaker HLA affinities of peptides containing driver mutations are unrelated to MHC genotypes

Assuming that differential HLA affinities between observed and unobserved mutations are indeed independent on the underlying MHC genotype, the affinity difference is still expected when all studied patients would have the same genotype. To test this hypothesis, we generated an *in silico* set of 10,295 virtual patients (VPs), each with a prototypical MHC-I genotype (using the most common alleles in a Caucasian population [14]; Fig 2A). Mutations were randomly attributed to each VP according to the mutation probabilities derived from the 688 driver mutation frequencies (e.g. the probability of a *BRAF* V600E mutation was 0.057, resulting in 550 VPs with this mutation). A comparison of the observed with the unobserved mutations in this set of VPs resulted in a similar difference as in the real patients (median PHBR 2.02 versus 1.54 respectively, $P$ = 1.09e-121; Fig 2B), confirming that this difference is indeed genotype-independent. Similar conclusions were obtained when affinities were calculated for a prototypical MHC-II genotype (see *Methods*; $P$ = 2.78e-74; Fig 2A and 2C).

As the only common denominator between both approaches (real patient versus VP data) is the mutation frequency, our results suggest that frequently occurring driver mutations are

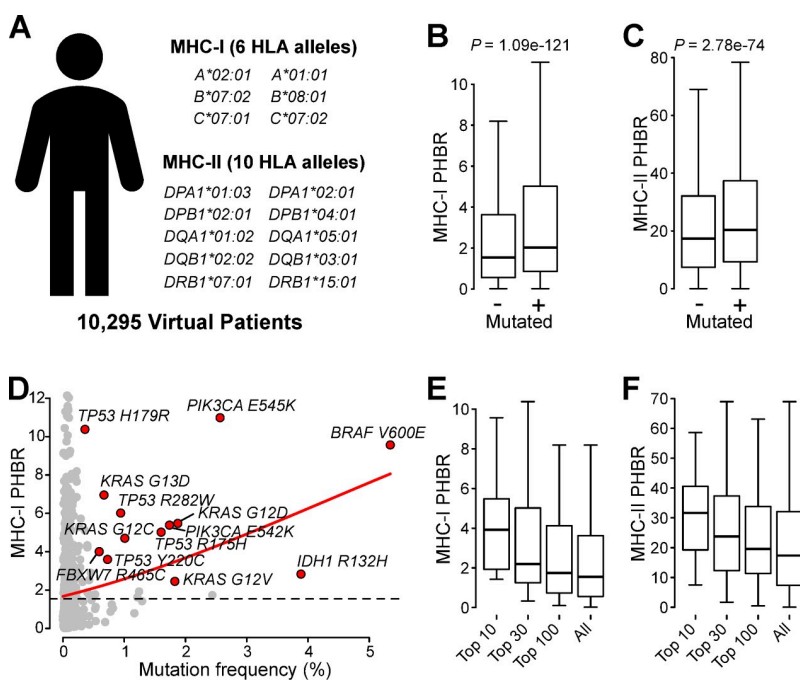

**Fig 2. HLA isogenotypic virtual patient analysis.** (A) Somatic driver mutations were simulated in a set of virtual patients, each characterized by the same prototypical MHC-I and MHC-II genotype. (B-C) Box plots comparing MHC-I (B) and MHC-II (C) PHBR values between observed (+) and unobserved (-) mutations. (D) Scatter plot showing correlation between mutation frequency and MHC-I PHBR values. Thirteen weak HLA affinity mutations, as identified in Fig 3, are highlighted in red and labelled. Loess regression line shown in red. Median PHBR value from all driver mutations indicated by dashed line. (E-F) Box plots comparing MHC-I (E) and MHC-II (F) PHBR values for 10/30/100 most frequent mutations as indicated. Box plots indicate median values and lower/upper quartiles with whiskers extending to 1.5 times the interquartile range. P values calculated using Wilcoxon rank-sum test.

characterized by higher PHBR values (lower HLA affinities). We could indeed confirm that the 10 most frequent mutations, which represent 25.1% (2451/9779) of all mutations, had the highest PHBR values (median PHBR 3.93 versus 1.54 for the complete dataset; Fig 2D and 2E). Similar results were obtained when PHBR scores were calculated considering the TCGA patient-specific MHC-I genotypes (S2 Fig) or based on the prototypical MHC-II genotype (Fig 2F). An orthogonal analysis using genomic data from a diverse set of 781 cancer cell lines yielded analogous results: higher PHBR scores for the most frequent mutations (median PHBR 3.36 for the top 10 versus 1.73 for the complete dataset), increased odds ratio using the within-patient model (1.24, P = 1.9e-13) and an effect that was also observed when using the prototypical MHC genotype (S3 Fig).

We next determined whether the observed PHBR differences were attributable to the specific composition of the prototypical MHC genotype or rather represent a more general HLA allele property. When the underlying MHC-I genotype was altered to the 2 most frequent HLA-A, HLA-B and HLA-C alleles from 5 other populations (European, American, South-Asian, East-Asian or African), PHBR values were similarly higher for observed than unobserved mutations (S4A Fig). Also considering HLA alleles with rare population frequencies did not change our conclusions (S4B Fig), suggesting that the PHBR difference is related to a general property of the individual HLA alleles. Indeed, an allele-based logistic regression analysis (within-patient model) on 193 different alleles confirmed that the ORs were significantly above 1 (5% FDR) for the majority (163/193, 84.5%) of all analyzed alleles. Notably, differences

were observed between the HLA genes, with 100% of HLA-C alleles, 87.4% of HLA-B alleles and 67.9% of HLA-A alleles having ORs above 1 (S5 Fig).

## Identification of 13 common driver mutations with high PHBR values

To identify the individual driver mutations responsible for the PHBR differences, we followed a "leave-one-out" approach where we reanalyzed the VP data by excluding the driver mutations one by one. To quantify the effect of each mutation, we derived the OR from the within-patient logistic regression model (Fig 1D) and calculated the percentage of OR reduction when the mutation was excluded from the dataset. The largest decrease in OR was observed when *BRAF* V600E was excluded from the analysis (33.4%, OR decreasing from 1.20 to 1.13), followed by *PIK3CA E545K* (16.5%) and *KRAS G12D* (7.2%; Fig 3). From all 688 mutations, only 13 mutations in 6 different genes were identified that reduced the OR with minimal 1%: *BRAF* (*V600E*), *IDH1* (*R132H*), *TP53* (*R175H, R282W, Y220C, H179R*), *KRAS* (*G12D, G12C, G13D, G12V*), *PIK3CA* (*E545K, E542K*) and *FBXW7* (*R465C*). These 13 mutations were all characterized by high mutation frequencies (all in the top 30 and 7 in the top 10 of the complete dataset; Fig 2D). When these 13 mutations were cumulatively excluded from the dataset, the correlation between PHBR and mutation probability disappeared (OR 0.99, $P = 0.32$), suggesting they were solely responsible for the association (Fig 3). The genomic regions containing these 13 mutations (mutated position +/- 10 base pairs) tended to be better conserved compared to the regions containing other driver mutations, as measured using the evolutionary conservation scores phyloP (median 5.97 vs. 4.88, $P = 0.023$, Wilcoxon rank-sum test) and phastCons (median 0.95 vs. 0.91, $P = 0.17$; S6 Fig).

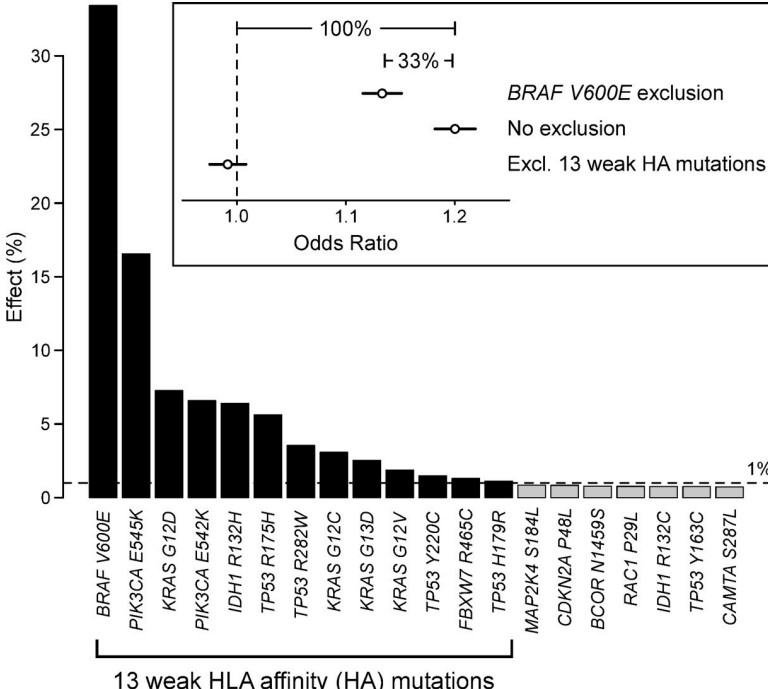

**Fig 3. Identification of 13 common driver mutations with weak HLA affinity.** Bar plot shows the effect size (%) of the 20 driver mutations in the dataset with the largest effect. This effect was determined by comparing a driver mutation's odds ratio (logistic regression, within-patient model) before and after exclusion, as illustrated in the inset for *BRAF V600E*. Mutations with an effect size larger than 1% (dashed line, dark bars) were considered weak HLA affinity mutations. Exclusion of these 13 mutations resulted in a complete removal of any effect (inset).

Repeating the leave-one-out analysis on the TCGA dataset directly (considering each patient-specific MHC-I genotype) confirmed the contribution of 11 of the 13 identified mutations to the PHBR difference, with another one just below the detection threshold of 1% (*KRAS G12V*, 0.92%; S2C Fig). PHBR scores were always similar between patients with observed and unobserved driver mutations, confirming the absence of any detectable selection pressure on the genotype level (S2D Fig).

## Common driver substitutions lead to weaker HLA affinities

While our results demonstrate higher PHBR values in 13 common oncogenic mutations, they do not explain the reason for this association. To understand the common denominator between the mutations' PHBR and oncogenic properties, we first examined whether a driver substitution itself could increase the PHBR value. Therefore, we calculated the PHBR values from the wild type peptides (i.e. containing the non-mutated, reference amino acid) and subtracted these values from the PHBR values derived from the mutated peptides. From the 13 identified mutations, we found the largest resulting ΔPHBR for *BRAF V600E* (ΔPHBR = 7.66), followed by *TP53 Y220C* (ΔPHBR = 3.51; Fig 4A and 4B).

To gain insight in the influence of different substitutions on HLA binding affinities, we simulated 15,000 random substitutions and calculated the ΔPHBR values for each substitution. Strikingly, from all 150 possible amino acid substitutions in the human genome, Val/Glu is the substitution resulting in the highest ΔPHBR values (median ΔPHBR = 4.48; Fig 4C). In general, we observed the highest ΔPHBR for substitutions where the mutated amino acid was either Asp (median ΔPHBR = 2.41) or Glu (median ΔPHBR = 1.76). Asp and Glu are both negatively charged amino acids that are known to mimic phosphorylation through their carboxyl groups, resulting in constitutive activation of oncogenes such as *BRAF* [23,24]. As charged amino acids are expected to decrease a peptides' HLA affinity (and vice versa for hydrophobic amino acids) [20], a phospho-mimicking mutation such as *BRAF V600E* is both oncogenic and at the same time increases PHBR values.

From the random substitution matrix shown in Fig 4C, it also becomes apparent that substitutions that involve (wild type) Tyr are characterized by the largest ΔPHBR values (median ΔPHBR = 2.17). These relatively high ΔPHBR values are in line with the long-known anchor function of Tyr in antigenic peptides binding in the MHC pocket (i.e. removing Tyr results in weaker binding and hence higher PHBR) [25]. At the same time, Tyr residues are important beta-sandwich stabilizers in secondary protein structures, implying that Tyr-involving mutations can result in protein dysfunction and hence oncogenic effects in tumor suppressor genes, as has been demonstrated for *TP53 Y220C* [26,27]. In line with these findings, we also observed large functional impact scores for these mutations (median PolyPhen-2 0.97; Fig 4C).

## Glycine residues surrounding *KRAS* mutations result in weaker HLA affinities

While substitutions like Val/Glu and Tyr/Cys explain the weaker HLA affinities found in peptides containing *BRAF V600E* or *TP53 Y220C* driver mutations, this cannot be the sole mechanism. Indeed, ΔPHBR is close to zero or even negative for other driver genes (Fig 4A). Furthermore, when focusing on PHBR values from wild type peptides only, the PHBR score difference between observed (median PHBR 1.91) and unobserved (median PHBR 1.51) mutations is still present ($P$ = 1.1e-34; Fig 5A), corresponding to 49.6% of the overall effect size as observed with the logistic regression analysis (OR = 1.10, $P$ = 8.42e-36; Fig 5B). These results suggest that other peptide properties than the substitution itself are at least partially responsible for the increased PHBR values. Based on our earlier observed association between mutation

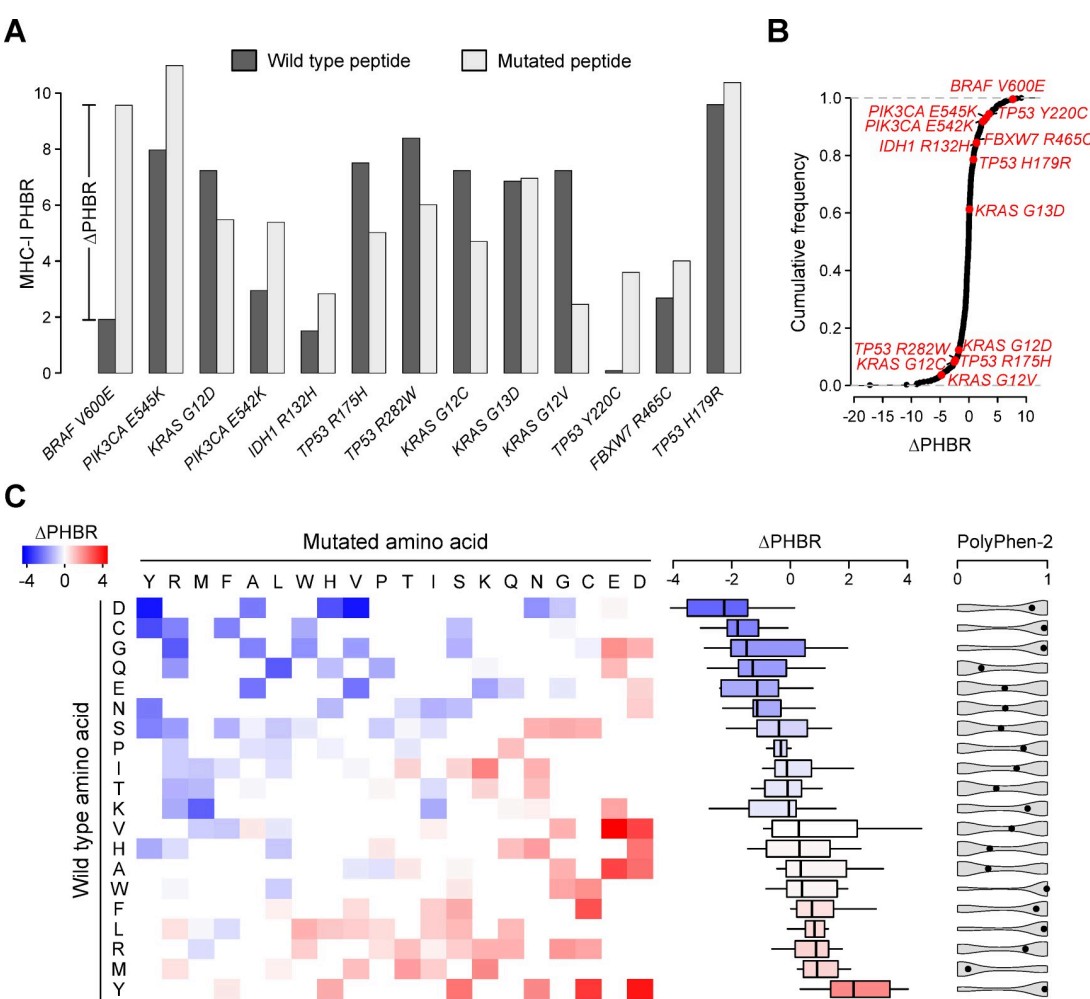

**Fig 4. Differential PHBR analysis.** (A) Bar plots comparing MHC-I PHBR values between wild type (PHBR_wt) and mutated peptides (PHBR), translated from the 13 identified weak HLA affinity driver mutations. ΔPHBR is defined as the difference between PHBR and PHBR_wt as illustrated for *BRAF V600E*. (B) ECDF plot for all driver mutations in the dataset with the weak HLA affinity mutations indicated and labelled in red. (C) Heatmap showing the expected ΔPHBR for each potential substitution in the human genome, based on 15,000 random simulations. Color key shown on top left. Rows and columns are ranked on median PHBR values as indicated by the box plots on the right of the heatmap. Violin plots show PolyPhen-2 functional impact scores for the corresponding substitutions. Box plots indicate median values and lower/upper quartiles with whiskers extending to 1.5 times the interquartile range.

sequence contexts, amino acid codons and HLA affinities [20], we hypothesized that the mutated peptides were enriched for HLA affinity decreasing amino acids.

As a first, general approach we checked for an association between the percentage of each amino acid in all 38 PHBR-determining peptides (see *Methods* and S1 Fig) and the mutation frequency. A positive correlation was observed for Gly (Pearson's r = 0.16, *P* = 1.46e-5; Fig 5C), but not for any other amino acid (at 5% FDR). These data suggest that frequent driver mutations have higher average Gly contents. However, Gly is a hydrophobic amino acid and this class of amino acids has been shown to result in higher HLA affinities, which is seemingly contradictive with the increased PHBRs (i.e. lower affinities). To understand this apparent inconsistency, we calculated the PHBR values from the peptides translated from a random set of somatic mutations and determined the influence of increasing the number of each individual amino acid on the PHBR value using a linear regression model. This regression analysis

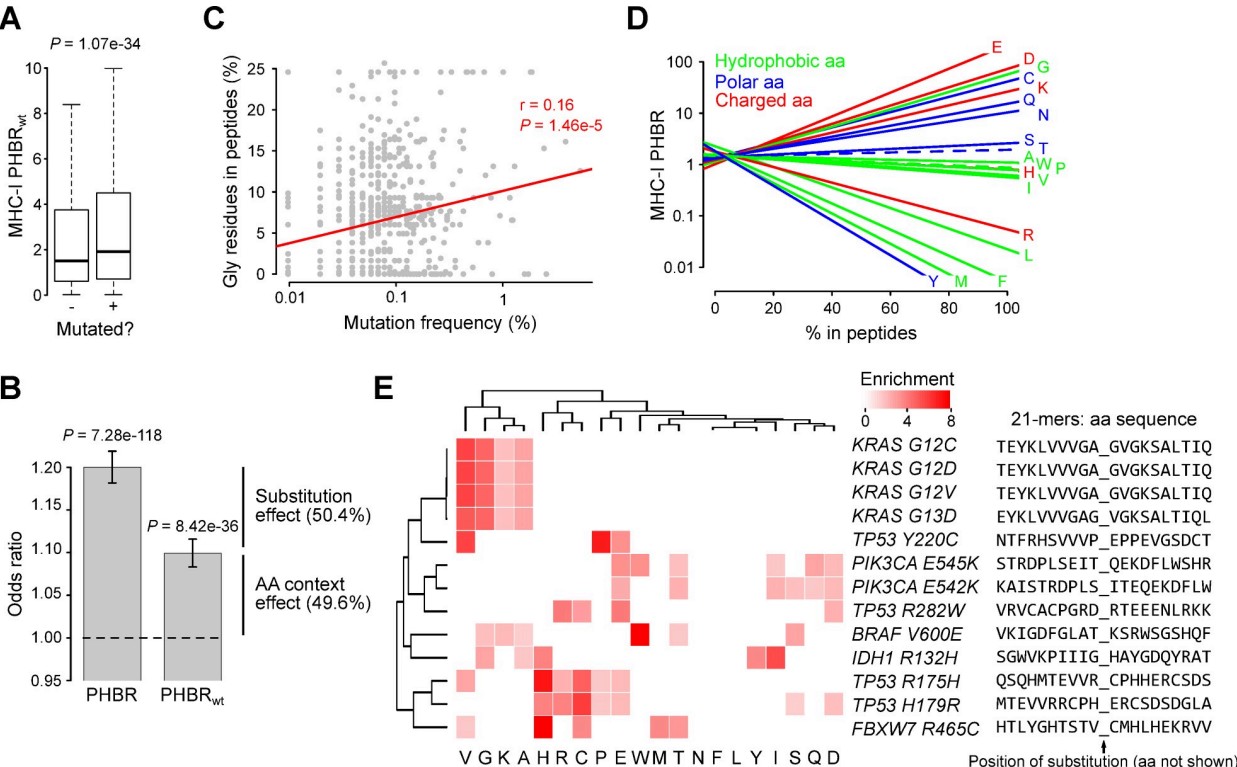

**Fig 5. Amino acid sequence context analysis.** (A) Comparison of PHBR$_{wt}$ values between observed (+) and unobserved (-) mutations. Box plots indicate median values and lower/upper quartiles with whiskers extending to 1.5 times the interquartile range. *P* values calculated using Wilcoxon rank-sum test. (B) Within-patient logistic regression analysis between PHBR or PHBR$_{wt}$ (as indicated) and mutation status. Bar plot shows odds ratios with 95% confidence intervals. (C) Correlation plot showing the percentage of Gly residues (in the 38 peptides underlying the MHC-I PHBR calculation, see *Methods*) as a function of the mutation frequency (logarithmic scale). Linear regression line shown in red. Pearson's correlation coefficient and *P* value indicated. (D) Linear regression lines indicating the correlation between the percentage of different amino acids and corresponding MHC-I PHBR values (logarithmic scale), based on 15,000 random peptides. Amino acid classes shown in different colors as indicated. Non-significant correlations (defined at 5% FDR) shown by dashed lines. (E) Heatmap showing amino acid enrichments for the 13 identified mutations. Amino acids indicated on the bottom and mutations (including perimutational amino acid sequence) on the right of the heatmap. Amino acid enrichment calculated using Fisher's exact test and only enrichments below 5% FDR are shown. Color key shown on top right.

demonstrates a positive correlation between the number of Gly residues and PHBR (*P* = 6.72e-140), which is exactly opposite to other hydrophobic amino acids and of an effect size similar to the charged amino acids Glu and Asp (Fig 5D).

To determine which mutations are responsible for this Gly correlation, we performed an amino acid enrichment analysis for each mutation-associated set of peptides. This analysis indicated that the effect was mainly attributed to a high Gly content in the 4 *KRAS* mutated peptides (all 4.7x enriched, *P* = 1.14e-27, Fisher's exact test; Fig 5E). The *KRAS* domain containing these 4 mutations is indeed characterized by a glycine-rich neighborhood (Gly at positions 10, 12, 13 and 15) and this has been functionally associated to GTP binding and activation properties [28,29]. In conclusion, our results show how *KRAS* activating mutations target a glycine-rich protein domain, resulting in the translation to low HLA affinity peptides.

Our amino acid enrichment analysis also showed higher-than-expected numbers of His and Cys residues in the immediate neighborhood of *TP53 R175H* and *H179R* (Cys 4.8x and 5.9x enriched for *R175H* and *H179R* respectively, *P* = 1.23e-21 and *P* = 8.79e-16; Fig 5E). Both amino acids are important for *TP53* interactions with zinc and interruption of this interaction has been shown to result in *TP53* instability and hence oncogenicity [26,30]. The association

with higher PHBR is thus explained by the strong PHBR-increasing effect of Cys residues (Fig 5D). Similarly, we also observed amino acid enrichments that could explain the correlation between oncogenicity and high PHBRs for the other mutations, although the precise mechanism was not always clear. In this regard it is worth mentioning Glu, Asp and Gln enrichments in the *PIK3CA E545K* and *E542K* peptides. These charged amino acids are all characterized by strong PHBR-increasing properties (Fig 5D) and have been shown important for interactions between the *PIK3CA*-encoded catalytic subunit and *PIK3R1*-encoded regulatory subunit of phosphoinositide 3-kinase alpha [31].

## Discussion

Neoantigens are small immunogenic peptides that are generated from mutated proteins by the antigen processing and presenting machinery (APPM), followed by binding to HLA molecules from the MHC complex. Neoantigen prediction from genome sequencing data is challenging. While the best results have been obtained using algorithms that predict the HLA binding capacity, it is noteworthy that, even if this binding prediction itself is highly accurate, HLA binding does not imply that the neoantigen will actually be generated by the APPM. Despite these limitations, HLA affinity predictions have been used extensively during the past 5 years to study immunogenic selection, often under the naïve assumption that these affinities have (near) 100% prediction accuracy and driven by enthusiasm about seemingly spectacular findings related to immunoediting in different cancer types. We have recently reanalyzed these data and demonstrated that results were strongly biased by improper usage of mutational signatures, genotype miscallings and related problems [20].

Our previous analysis also indicated weaker HLA binding affinities in a set of *in silico* generated peptides generated from cancer driver genes, questioning the interpretation from 2 other studies that attributed similar HLA affinity alterations to MHC-I and MHC-II genotype restrictions of the oncogenic mutational landscape (i.e. driver mutations only occur when they are not presentable by the underlying MHC genotype) [21,22]. Although this interpretation follows biological intuition, methodological concerns have been raised [32]. To provide conclusive evidence that the earlier findings are indeed unrelated to the underlying MHC genotype, we generated a set of *in silico* virtual patients, based on a single prototypical genotype, and demonstrated that the correlation between HLA affinities and mutation probability was caused by weak binding affinities of peptides translated from 13 common driver mutations in 6 different genes.

We demonstrated how several known oncogenic mechanisms that cause constitutive activation of oncogenes (e.g. phospho-mimicking) or inactivation of tumor suppressor genes (e.g. destabilization of secondary protein structures) concomitantly result in either a direct lowering of HLA affinities (e.g. *BRAF V600E, TP53 Y220C*) or occur in protein regions that are characterized by amino acid residues with weak HLA binding properties (e.g. *KRAS G12C/D/V* and *G13D, PIK3CA E542K* and *E545K, TP53 R175H* and *H179R*) (Fig 6). We are aware that other, possibly yet unidentified, oncogenic mechanisms exist that might be associated to lower HLA binding capacities. Relatedly, our background HLA affinity analysis was focused on single amino acid substitutions only and more complex interactions (e.g. between substitutions and specific amino acid sequences) might have been missed. This is exemplified by the positive ΔPHBR values for *PIK3CA E542K* and *E545K*, which cannot be explained by single E/K substitution, as this is expected to decrease ΔPHBR values (Fig 5D). Further, while the virtual patient approach is a simple and powerful method to rule out somatic selection effects at the MHC genotype level and explore the nature of alternative effects, some correlations between amino acids and HLA affinities might be different for other HLA alleles.

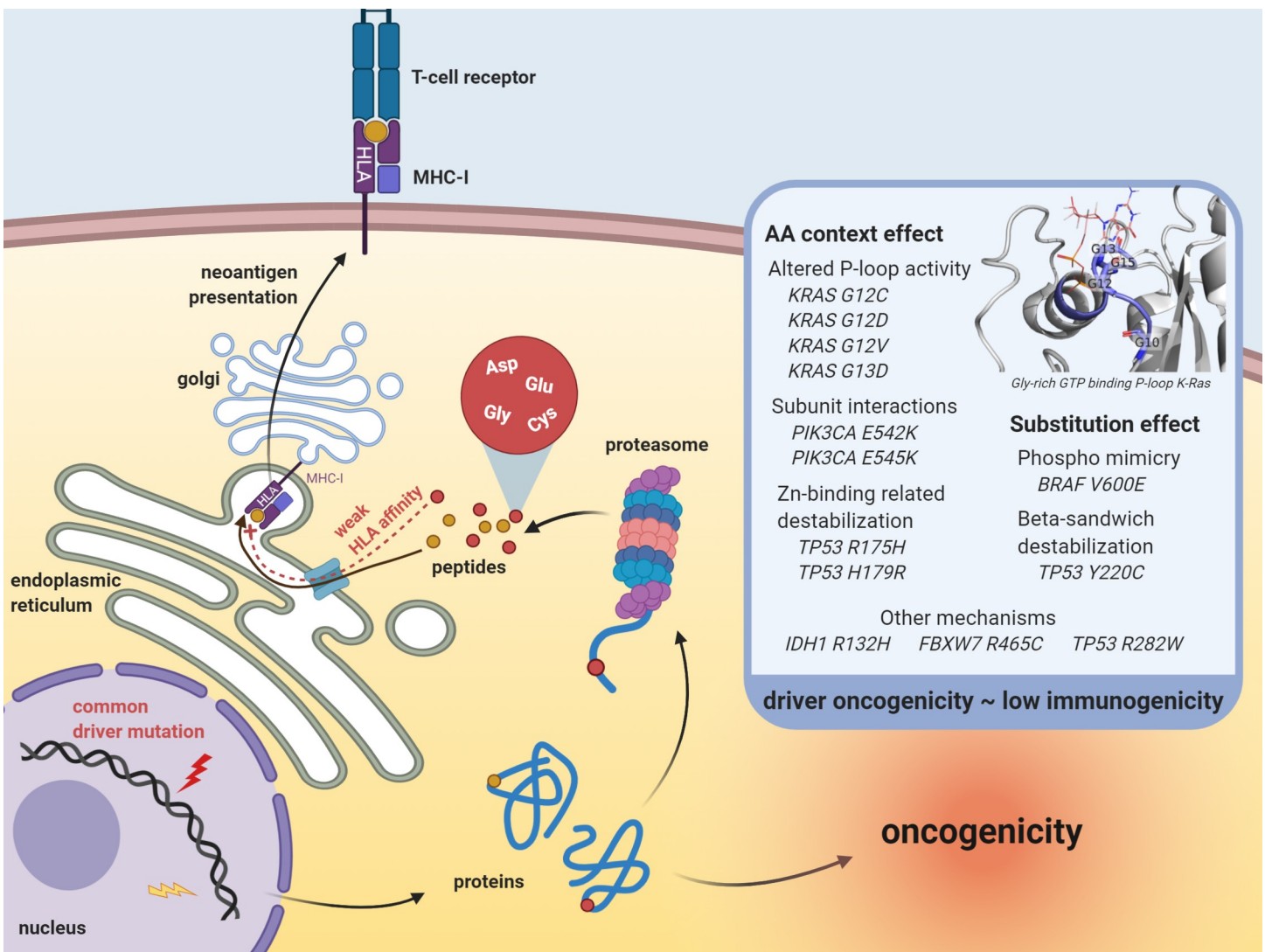

**Fig 6. Common driver mutations are associated with weak HLA affinities.** Summary of the findings from this study. Thirteen common driver mutations are associated with weak HLA affinities of their translated peptides, either through the substitution itself (substitution effect) or by their occurrence in protein domains characterized by amino acids with a negative influence on HLA binding (AA context effect). Protein model in the inset was created in PyMol and represents the glycine-rich GTP binding P-loop in K-Ras (PDB accession nr. 4OBE). Figure created with BioRender.com.

We showed that the weaker HLA affinities in peptides translated from these 13 driver mutations were a general HLA allele property. It remains an open question whether these altered affinities are a simple epiphenomenon, resulting from the more efficient oncogenic mechanisms, or, alternatively, are related to differing germline evolutionary pressures (e.g. related to host-pathogen coevolution [33]). While a trend was observed towards better evolutionary conservation for the genomic regions containing the 13 mutations, any putative causal relationship remains to be determined.

Our results point to alternative explanations for earlier perceived HLA affinity differences but do not exclude that existence of actual immune selection signals on neoantigen forming mutations. They do illustrate how HLA affinity predictions are easily misinterpreted as immunogenic selection signals. Given the low prediction accuracies for true neoantigen formation and the many sources of bias (mutational signatures, gene expression, hot spots, . . .), we argue

for a critical re-evaluation and interpretation of HLA affinity predictions when studying immunoediting and immunogenic selection.

## Methods

### Somatic mutation data

MC3-called whole exome sequencing (WES) mutation annotation format (maf) files from The Cancer Genome Atlas (TCGA) were downloaded from Genomic Data Commons (https://api.gdc.cancer.gov/data/1c8cfe5f-e52d-41ba-94da-f15ea1337efc). These data include all 33 available cancer types from 10,295 patients. Main analyses were performed on the complete, pan cancer dataset. Individual cancer type analyses were performed on 25 different cancer types that contained minimal 100 samples (see Fig 1B and 1E for an overview).

Recurrent driver mutations (688 in total) were selected by applying the following 3 filtering criteria on the pan cancer dataset: 1) Variant: missense mutation, 2) Gene present in Cancer Gene Census (CGC) and 3) Minimally 5 recurrent mutations in the dataset. CGC (v91) information was downloaded from COSMIC (https://cancer.sanger.ac.uk/cosmic/download).

A binary patient (n = 10,295) x driver mutation (n = 688) mutation matrix was then created from the data where each cell contains a binary value, indicating whether the mutation was observed for the corresponding sample or not (0: no observed mutation; 1: observed mutations; Fig 1A).

### HLA genotyping

TCGA MHC-I genotypes were obtained from an earlier study [20]. To determine the 6 HLA alleles (2 HLA-A alleles, 2 HLA-B alleles and 2 HLA-C alleles) that composed the MHC-I genotype, HLA typing was performed on WES normal bam files from all available TCGA samples using Polysolver [34]. Polysolver was run using default settings and without setting prior population probabilities, resulting in the successful genotyping of 8,969 TCGA samples. HLA typing was validated by comparison of the derived allele frequencies with a healthy blood donor population, downloaded from Allele frequency net [14].

### HLA affinity predictions and PHBR scores

HLA affinities were predicted using NetMHCpan 4.0 for MHC-I and NetMHCIIpan 3.2 for MHC-II [15,16]. We first determined the amino acid sequence of the 29-meric peptide containing the mutation (from 14 upstream to 14 downstream amino acids) using EnsemblDB v86 (S1 Fig). HLA affinities were then calculated for 38 possible 8–11 mers for MHC-I and 15 possible 15-mers for MHC-II, and for 6 MHC-I alleles (2 copies of HLA-A, HLA-B and HLA-C genes) and 10 MHC-II alleles (2 copies of HLA-DPA, HLA-DPB, HLA-DQA, HLA-DQB and HLA-DRB genes) respectively. HLA affinities were quantified using rank-based scores (lower rank implies stronger affinity) and for each mutation, the Patient Best Rank (PBR) score (i.e. minimum value) was determined for the 6 MHC-I alleles and 10 MHC-II alleles. The harmonic mean was then calculated from the allelic PBR scores, resulting in the MHC-I and MHC-II Patient Harmonic Best Rank (PHBR) score as an indicator of overall HLA affinity (S1 Fig) [22]. PHBR scores were also calculated for the wild type peptides (PHBR$_{wt}$). ΔPHBR was defined as the difference between a PHBR and PHBR$_{wt}$.

PHBR scores were determined for both the TCGA patient-specific HLA MHC-I genotypes and prototypical MHC-I and MHC-II genotypes. These prototypical genotypes are composed of the 2 most common alleles for each HLA gene, as derived from a healthy Caucasian US blood donor population from Allele frequency net [14] (MHC-I: HLA A02:01, A01:01, B07:02,

B08:01, C07:01, and C07:02; MHC-II: HLA DPA10103, DPA10201, DPB10201, DPB10401, DQA10102, DQA10501, DQB10301, DQB10202, DRB10701 and DRB11501). PHBR scores were then put in an HLA affinity matrix, corresponding to the mutation matrix (Fig 1A).

Population-specific genotypes were derived from an earlier study that called HLA alleles from 1000 genomes project NGS data [35]. These data were downloaded from http://ftp.1000genomes.ebi.ac.uk/vol1/ftp/data_collections/HLA_types/20181129_HLA_types_full_1000_Genomes_Project_panel.txt After exclusion of 116 samples with ambiguous or absent HLA calls, the most frequent HLA-A, HLA-B and HLA-C alleles were derived for each super-population: AFR, African (n = 674); AMR, Ad Mixed American (n = 359); EAS, East Asian (n = 511); EUR, European (n = 512); SAS, South Asian (n = 521).

## Analysis of mutation-PHBR correlations

To determine PHBR score differences between observed and unobserved mutations, 2 different approaches were used: Wilcoxon rank-sum test and logistic regression. For the logistic regression analysis, we correlated the log PHBR (observed variable) to the corresponding mutation status (binary response variable), considering 2 random effect models: 1) a within-mutation model where the random effects were determined by the different mutation frequencies and 2) a within-patient model where the random effects were determined by the different mutation loads per patient. These models were fitted using the *glmer* function from the R package *lme4* [36]. The correlation was then quantified using odds ratios (ORs), with ORs higher than 1 indicating positive correlations, i.e. higher PHBR scores for observed mutations. 95% confidence intervals were obtained using the Wald method (R *stats* package, *confint* function).

## Virtual patients

A set of 10,295 isogenotypic virtual patients was created *in silico*, each containing the prototypical MHC-I and MHC-II genotype and with mutations based on the mutation driver frequencies from the real dataset. These mutations were attributed to each patient by randomly sampling from a binary 0/1 vector with replacement and with probabilities given by the different mutation frequencies.

## Leave-one-out analysis and identification of weak HLA affinity mutations

To identify the mutations responsible for the PHBR-mutation correlations, the within-patient logistic regression model was run iteratively, each time after exclusion of 1 mutation. The effect of each mutation on the PHBR-mutation correlation was then quantified by normalization of the difference between the obtained OR with the baseline OR (no exclusion) to the maximal effect (baseline OR– 1; Fig 3). Weak HLA affinity mutations were identified as mutations with normalized values higher than 1% (13 in total).

## Simulations of substitutions

All point mutations in the human genome were simulated, resulting in 150 possible missense substitutions. We randomly sampled 100 substitutions from each substitution type and generated a dataset containing information for 15,000 substitutions in total. Annotations and Poly-Phen-2 functional impact scores were derived using ANNOVAR [37]. Peptide information was added using EnsemblDB and HLA affinities were calculated as described higher.

A reference amino acid (n = 20) x mutated amino acid (n = 20) substitution matrix was created with each cell containing the median ΔPHBR from all 100 substitutions (or NA if the substitution didn't occur in the genome). A heatmap was generated using the *heatmap.2* function

from the R package *gplots* with row and column ranking based on the median values of the corresponding rows and columns of the substitution matrix.

To determine whether and how the number of amino acids determined the PHBR scores, a linear regression analysis was performed between the percentage of the number of each amino acid in the 38 PHBR-determining peptides (observed variable) and the log PHBR (response variable). Amino acids were grouped in 3 classes: hydrophobic (Gly, Ala, Pro, Val, Leu, Iso, Met, Trp and Phe), polar (Ser, Thr, Tyr, Asn, Gln and Cys) and charged (Lys, Arg, His, Asp and Glu).

### Amino acid enrichment analysis

To search for enrichments in the amino acids surrounding each of the 13 identified weak HLA affinity mutations, the number of amino acids was counted for each mutation's 38 PHBR-determining peptides (excluding the mutated amino acid). A contingency table was then created, comparing for each mutation the amino acid's proportion to the background proportion, as derived from the complete dataset. Enrichment was determined using Fisher's exact test. A heatmap was generated using the *heatmap.2* function with default settings.

### Evolutionary conservation analysis

To determine and compare evolutionary conservation between different sets of mutations, phastCons100way and phyloP100way scores were calculated for a 21 base pair genomic region extending from 10 base pairs upstream to 10 base pairs downstream of each somatic mutation. Scores were derived using the *GenomicScores* R package [38].

### Cancer cell line data

Cancer cell line data were derived from the Cancer Cell Line Encyclopedia (CCLE) [39]. Mutation data and sample information was downloaded from DepMap (release 20Q4) at https://depmap.org/portal/. HLA calls were derived from an earlier study [40].

### Data processing and statistical analysis

The R statistical package (v3.6) was used for all data processing and statistical analysis. Details on the statistical tests used in this study are reported in the respective sections. *P* values less than 0.05 were considered significant for individual tests. For multiple comparisons, false discovery rate (FDR) corrections were performed using the Benjamini-Hochberg method [41].

## Supporting information

**S1 Fig. MHC-I and MHC-II PHBR calculations.** Perimutational amino acid sequences (21-mers for MHC-I, 29-mers for MHC-II) were derived using EnsemblDb. Based on these sequences, all 8 to 11-mers (MHC-I, top) and 15-mers (MHC-II, bottom) containing the mutation were determined as illustrated. For each peptide and MHC-I/MHC-II allele, the HLA affinities were predicted and quantified using a rank-based score. The Patient Best Rank (PBR) was then obtained from the best binding (lowest) ranks. Finally, the Patient Harmonic Best Rank (PHBR) was calculated from the PBRs. Numbers were derived from *BRAF V600E* (prototypical MHC genotypes, see *Methods*) and given for illustration purposes. Note that MHC-II DP and DQ genes form heterodimers between alpha and beta subunits. (PDF)

**S2 Fig. TCGA real patient analysis.** PHBR scores were calculated considering TCGA MHC-I genotypes (as in Fig 1). (A) Scatter plot showing the correlation between mutation frequency

and median PHBR values. Thirteen weak HLA affinity mutations, as identified in Fig 3, are highlighted in red and labelled. Loess regression line shown in red. Median PHBR value from all 688 analyzed driver genes indicated by dashed line. (B) Box plot comparing PHBR values for the 10/30/100 most frequent mutations as indicated. (C) Bar plot showing the effect size (%) of the 20 driver mutations in the dataset with the largest effect. See Fig 3 for details on effect size calculations. Bars corresponding to the 13 weak HLA affinity mutations from the main analysis are colored in black. (D) Comparison of PHBR scores between patients with observed and unobserved mutations. Box plots show scores for the 13 weak HLA affinity mutations identified in the main analysis (left) and the 3 additional mutations with minimal effect size of 1% as identified in panel C (right). Box plots indicate median values and lower/ upper quartiles with whiskers extending to 1.5 times the interquartile range. *P* values calculated using Wilcoxon rank-sum test.
(PDF)

**S3 Fig. Cancer cell line analysis.** PHBR scores were calculated for all hot spot driver muta-tions in cancer cells from the Cancer Cell Line Encyclopedia (CCLE), for which HLA allele calls were available. Analysis in (A-E) are based on sample-specific HLA alleles, while (F-G) are based on the prototypical MHC-I genotype. (A) Pie chart showing the different primary tumors where the cell lines were derived from. (B) Box plots compare PHBR values between observed (+) and unobserved (-) mutations. (C) Logistic regression analysis between log PHBR (observed variable) and mutation status (response variable). Analysis was performed using both the within-patient and within-mutation regression model as indicated (see main analysis for details). (D) Scatter plot showing correlation between mutation frequency and median PHBR values. Thirteen weak HLA affinity mutations, as identified in Fig 3, are highlighted in red and labelled. Loess regression line shown in red. Median PHBR value from all 688 analyzed driver genes indicated by dashed line. (E) Box plot comparing PHBR values for 10/30/100 most frequent mutations as indicated. (F-G) Similar as panels (B) and (C) but with PHBR values calculated based on the prototypical MHC-I genotypes. Box plots indicate median values and lower/upper quartiles with whiskers extending to 1.5 times the interquartile range. *P* values calculated using Wilcoxon rank-sum test.
(PDF)

**S4 Fig. MHC-I PHBR analysis using common and rare HLA alleles from different popula-tions.** PHBR scores were calculated considering different MHC-genotypes as indicated. Box plots compare PHBR scores between observed (+) and unobserved (-) mutations. Muta-tions were derived from the virtual patient mutation matrix used in the main analysis. (A) MHC-I genotype composed of the 2 most frequent HLA-A, HLA-B and HLA-C alleles for dif-ferent populations from the 1000 genomes project. (B) MHC-I genotype composed of 2 ran-dom HLA-A, HLA-B and HLA-C alleles that were never observed in the indicated population. Box plots show median values and lower/upper quartiles with whiskers extending to 1.5 times the interquartile range. *P* values calculated using Wilcoxon rank-sum test. AFR, African; AMR, Ad Mixed American; EAS, East Asian; EUR, European; SAS, South Asian.
(PDF)

**S5 Fig. HLA allele-based analysis.** Patient Best Rank (PBR) scores were calculated for each of 193 different alleles that were called minimally once in the TCGA dataset. The within-patient logistic regression analysis was performed considering the mutation matrix used for the virtual patient main analysis. Plot shows odds ratios (OR) with 95% confidence intervals for each allele with HLA-A, HLA-B and HLA-C indicated by colors and alleles with highest/lowest OR labelled. Alleles with OR above/below 1 (determined at 5% FDR) with corresponding

frequencies are indicated below the plot.
(PDF)

**S6 Fig. Evolutionary conservation analysis.** Box plots compare MHC-I PHBR scores (A) and 2 evolutionary conservation scores (B, pastCons; C, phyloP) between different somatic mutation datasets. From left to right and as indicated: 500 random missense mutations that were never observed in the TCGA dataset; 500 random missense mutations that were not recurrent (occurred once in the TCGA dataset) and are not known as a Cancer Gene Census (CGC) gene; 500 recurrent missense mutations that are not known as CGC genes; 500 non-recurrent missense mutations in CGC genes; 688 recurrent missense mutations in CGC genes (the dataset used for the main analysis) that are further divided depending on the presence of the gene in the 13-gene weak HLA affinity mutation dataset identified in the main analysis. phastCons (B) and phyloP (C) were derived from multiple alignments between 100 vertebrate species and calculated for the genomic region extending +/- 10 base pairs from the mutation (21 base pairs in total). Box plots indicate median values and lower/upper quartiles with whiskers extending to 1.5 times the interquartile range. *P* values calculated using Wilcoxon rank-sum test.
(PDF)

## Acknowledgments

The results shown here are in whole or part based upon data generated by the TCGA Research Network: https://www.cancer.gov/tcga.

## Author Contributions

**Conceptualization:** Jimmy Van den Eynden.

**Data curation:** Arne Claeys, Tom Luijts, Jimmy Van den Eynden.

**Formal analysis:** Arne Claeys, Tom Luijts, Jimmy Van den Eynden.

**Funding acquisition:** Jimmy Van den Eynden.

**Investigation:** Arne Claeys, Jimmy Van den Eynden.

**Methodology:** Jimmy Van den Eynden.

**Project administration:** Jimmy Van den Eynden.

**Resources:** Arne Claeys, Jimmy Van den Eynden.

**Software:** Arne Claeys, Jimmy Van den Eynden.

**Supervision:** Jimmy Van den Eynden.

**Validation:** Jimmy Van den Eynden.

**Visualization:** Arne Claeys, Tom Luijts, Jimmy Van den Eynden.

**Writing – original draft:** Arne Claeys, Kathleen Marchal, Jimmy Van den Eynden.

**Writing – review & editing:** Arne Claeys, Tom Luijts, Kathleen Marchal, Jimmy Van den Eynden.

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
