## [Decision Letter · Decision Letter 0]

20 Dec 2020

Dear Dr Van den Eynden,

Thank you very much for submitting your Research Article entitled 'Low immunogenicity of common cancer hot spot mutations resulting in false immunogenic selection signals' to PLOS Genetics.

The manuscript was fully evaluated at the editorial level and by independent peer reviewers. The reviewers appreciated the attention to an important topic but identified some concerns that we ask you address in a revised manuscript

We therefore ask you to modify the manuscript according to the review recommendations. Your revisions should address the specific points made by each reviewer. Two of the reviewers noted that an analysis that included less common HLAs might also be informative, or HLAs drawn from an unrelated population study (rev 3). Reviewer 3 additionally suggested some analyses on the conservation of the mutations and surrounding sequence. 

[LINK]

Yours sincerely,

Paul G. Thomas

Guest Editor

PLOS Genetics

Peter McKinnon

Section Editor: Cancer Genetics

PLOS Genetics

Reviewer's Responses to Questions

**Comments to the Authors:**

Reviewer #1: This article is very interesting, well written and clearly argues for a critical re-evaluation and interpretation of predictions based on HLA-binding affinity when studying immunoediting. Like all papers that challenge a dogma, this paper could stir some controversy. Nonetheless, I found that the analyses were solid and yielded several nuggets of information on common driver mutations. In my opinion, this article should be of great importance to the fields of cancer genetics and cancer immunology.

Minor Point:

Line 59-60: What are the three regions that correspond to the six mentioned loci?

Reviewer #2: The paper by Claeys, Marchal and Van den Eynden addresses the tendency of known cancer driver mutations to not be very well presented by the MHC complex. The authors challenge the common argument that this is due to selection bias by the immune system. They first show that observing a specific driver mutation is unrelated to the specific HLA genotype of the individual, and indeed a few common mutations have low presentation leading to this effect. They then highlight some of these specific mutations and show why they decrease the peptides’ HLA affinity generally.

The paper is based on sound and thorough statistical analysis. The results address a current research question in a timely manner. The paper is convincing and interesting to researchers interested in the interaction of the immune system and cancer. My main comment is that the analysis in some places can be obfuscating and unintuitive, especially to non-statisticians. Some choices and assumptions are not explained clearly enough. I suggest making some of the claims and arguments more transparent. Specific points:

1) The main result was the association between the frequency of driver mutations and the PHBR score. It would be both interesting and helpful to show a scatter plot of these two quantities for the TCGA data, maybe on a log scale, highlighting the (13) mutations most contributing to the association. This would support the statistical analysis already done in a clear manner.

2) Most of the results were based on HLA binding analysis using a prototypical HLA genotype. While the authors did show that the low presentability of the mutations doesn’t seem to be HLA specific, it’s not clear if certain rarer choices of HLA alleles wouldn’t change this. Specifically I would be interested to see if the results for the weaker HLA affinity for the 13 driver mutations hold for different choices of HLA, maybe a random pick or based on different populations.

3) The 688 mutations used for the binary matrix were recurrent missense mutations, so all of the them were observed in at least some of the patients. Can the authors say anything about mutations which were really not observed, by simulating them and checking HLA affinity?

4) Line 169: why only 150 possible amino acid substitutions in the human?

5) Line 172: Do Glu and Gly suppose to be the same amino acid?

6) Line 191: is the d in “dPHBR” should be a delta sign?

7) Line 192: it is not clear how the substitution effect is neutralized by focusing on PHBR values from wild type peptides.

8) Line 202: what are the 38 PHBR-determining peptides and how are they determined?

9) Line 347: “This data” should be “These data” (data is plural)

Reviewer #3: see attached.

**Have all data underlying the figures and results presented in the manuscript been provided?**

Reviewer #1: Yes

Reviewer #2: Yes

Reviewer #3: Yes

PLOS authors have the option to publish the peer review history of their article (what does this mean?). If published, this will include your full peer review and any attached files.

Reviewer #1: No

Reviewer #2: **Yes: **Yuval Elhanati

Reviewer #3: No

---

## [Editor Report · Decision Letter 1]

13 Jan 2021

Dear Dr Van den Eynden,

We are pleased to inform you that your manuscript entitled "Low immunogenicity of common cancer hot spot mutations resulting in false immunogenic selection signals" has been editorially accepted for publication in PLOS Genetics. Congratulations!

Yours sincerely,

Paul G. Thomas

Guest Editor

PLOS Genetics

Peter McKinnon

Section Editor: Cancer Genetics

PLOS Genetics

Comments from the reviewers (if applicable):

**Data Deposition**

http://datadryad.org/submit?journalID=pgenetics&manu=PGENETICS-D-20-01663R1

**Press Queries**

---

## [Editor Report · Acceptance letter]

2 Feb 2021

PGENETICS-D-20-01663R1 

Low immunogenicity of common cancer hot spot mutations resulting in false immunogenic selection signals 

Dear Dr Van den Eynden, 

We are pleased to inform you that your manuscript entitled "Low immunogenicity of common cancer hot spot mutations resulting in false immunogenic selection signals" has been formally accepted for publication in PLOS Genetics! Your manuscript is now with our production department and you will be notified of the publication date in due course.

With kind regards,

Alice Ellingham

PLOS Genetics

On behalf of:
